# Trapping of Ag^+^ into a Perfect Six-Coordinated Environment: Structural Analysis, Quantum Chemical Calculations and Electrochemistry

**DOI:** 10.3390/molecules27206961

**Published:** 2022-10-17

**Authors:** Veronika I. Komlyagina, Nikolay F. Romashev, Vasily V. Kokovkin, Artem L. Gushchin, Enrico Benassi, Maxim N. Sokolov, Pavel A. Abramov

**Affiliations:** 1Nikolaev Institute of Inorganic Chemistry, Siberian Branch of Russian Academy of Sciences (SB RAS), 3 Akad. Lavrentiev Ave., 630090 Novosibirsk, Russia; 2Faculty of Natural Sciences, Novosibirsk State University, 2 Pirogova Str., 630090 Novosibirsk, Russia; 3Institute of Natural Sciences and Mathematics, Ural Federal University, 620002 Ekaterinburg, Russia

**Keywords:** polyoxomolybdates, Ag, dpp-bian, redox-active ligands, complexes, crystal structure, DFT calculations, cyclic voltammetry

## Abstract

Self-assembly of (Bu_4_N)_4_[β-Mo_8_O_26_], AgNO_3_, and 2-bis[(2,6-diisopropylphenyl)-imino]acenaphthene (dpp-bian) in DMF solution resulted in the (Bu_4_N)_2_[β-{Ag(dpp-bian)}_2_Mo_8_O_26_] (**1**) complex. The complex was characterized by single crystal X-ray diffraction (SCXRD), X-ray powder diffraction (XRPD), diffuse reflectance (DR), infrared spectroscopy (IR), and elemental analysis. Comprehensive SCXRD studies of the crystal structure show the presence of Ag^+^ in an uncommon coordination environment without a clear preference for Ag-N over Ag-O bonding. Quantum chemical calculations were performed to qualify the nature of the Ag-N/Ag-O interactions and to assign the electronic transitions observed in the UV–Vis absorption spectra. The electrochemical behavior of the complex combines POM and redox ligand signatures. Complex **1** demonstrates catalytic activity in the electrochemical reduction of CO_2_.

## 1. Introduction

Ag^+^/L/polyoxometalate hybrid organic–inorganic systems are attractive candidates for engineering multifunctional coordination networks. In this field, numerous works have been contributed by Cronin [1,2,3,4,5], Wang [6,7], Sun [8,9,10], Mac [11], and Niu et al. [12,13]. The versatility of Ag-based supramolecular building blocks opens almost limitless opportunities to create coordination networks of different topologies and compositions [14,15,16,17,18,19,20,21,22]. Complexes with redox-active ligands have been mostly ignored in this field. We believe, however, that the usage of such ligands would offer an advantage to construct new semiconductors with controllable topologies.

Bis(arylimino)acenaphthenes (BIANs) are redox-active, sterically bulky ligands, widely used as N,N-bidentate ligands in coordination chemistry and catalysis [23,24]. The key feature of BIANs as strong π-acceptor molecules is their ability to accept up to four electrons, which can be reversibly exchanged with the coordinated metal-triggering redox-based chemical processes [25,26,27,28,29,30,31,32,33,34,35]. Many complexes of late transition metals with BIANs have been reported [36,37,38,39,40,41]. However, Ag complexes with BIANs remain scarcely explored: There are only seven structurally characterized examples of Ag-BIAN complexes [42,43,44,45]. The first mention is found in the report by Rosa and co-workers, who reported the preparation, spectroscopic, and computational studies of [Ag(dpp-bian)_2_]BF_4_ and [Ag(tmp-bian)_2_]BF_4_ [42].

In our recent studies, we paid particular attention to the Ag^+^ coordination environment in novel [β-{AgL}_2_Mo_8_O_26_]^2−^ complexes, where the [β-Mo_8_O_26_]^4−^ polymolybdate acts as a ditopic doubly tetradentate ligand towards Ag^+^ [46,47,48]. These works focus on the Ag^+^ coordination behavior and changing of the coordination number (CN) initiated by the auxiliary ligand L. In all cases, we used monodentate ligands for L, and found typical coordinations with CN 3 and 4. Moreover, Ag^+^ can form long-range contact with an additional DMF molecule, formally increasing its CN [46]. This situation may be more complicated without any auxiliary ligand when different monomeric and polymeric species have been detected [47]. Concerning bidentate ligands, either rigid (e.g., BIANs) or flexible (e.g., bpy and its derivatives) diimines with redox nature can be used, which opens numerous ways to construct new coordination networks, with relevance in the field of coordination polymers preparation. In this work we report a new complex, *viz.* (Bu_4_N)_2_[β-{Ag(dpp-bian)}_2_Mo_8_O_26_], indicating a unique Ag coordination environment. The electrochemical activity of the complex toward CO_2_ reduction was studied.

## 2. Results and Discussion

### 2.1. Synthesis and Structural Features

Mixing (Bu_4_N)_4_[β-Mo_8_O_26_] and AgSO_3_CF_3_ in DMF results in the coordination of Ag^+^ with [β-Mo_8_O_26_]^4−^, leading to [{Ag(DMF)_x_}_2_Mo_8_O_26_]^2−^ (x = 1, 2) and other species [47]. The addition of dpp-bian to such a solution provided an orange precipitate. Brown block-shaped crystals of **1** suitable for X-ray diffraction analysis were obtained by diffusion of diethyl ether into a solution of **1** in DMF. The resulting crystalline complex **1** crystallized in monoclinic crystal system (*C*2/c space group) with the following unit cell parameters: a = 38.5675(16), b = 12.6898(6), c = 26.2540(11) Å, β = 116.858(1)° (Appendix A Appendix A). The main building blocks of the structure are hybrid [β-{Ag(dpp-bian)}_2_Mo_8_O_26_]^2−^ anions (Figure 1).

It should be noted that using other bians (e.g., tmp-bian, or *p*-Cl-bian) with less bulky substituents always provided the corresponding [Ag(bian)_2_]SO_3_CF_3_ complexes. The most proposed cause of this is in bulkiness of the substituents.

The main structural feature in this complex is the unusual coordination environment of Ag^+^ (Figure 2, Appendix A) consisting of four oxo ligands from polyoxomolybdate and two N-atoms from dpp-bian. The Ag-O bond distances ranged from 2.495 to 2.569 Å, while Ag-N distances were 2.386 and 2.461 Å. The asymmetric coordination of the dpp-bian ligand was likely caused by packing effects. The Ag^+^, therefore, resides inside a coordination environment that has six coordination bonds, as demonstrated by the quantum chemical calculations (*vide postea*). Unlike many other structures that involve multiple Ag···L contacts, the bond lengths vary in a relatively narrow range (0.18 Å) and cannot be divided into primary and secondary interactions. The coordination polyhedron of Ag^+^ is irregular, neither octahedral nor trigonal prismatic, but is similar to a monocapped octahedron, lacking a vertex in the capped face (Figure 3). The analysis with SHAPE (http://www.shapesoftware.com/ (accessed on 3 September 2022)) shows that this is a transition state between a pentagonal pyramid and a trigonal prism (Appendix A).

The sum of the ionic radii of Ag^+^ (1.15 Å, Shannon for the CN 6) and O^2−^ (1.35 Å, Shannon) is 2.50 Å. Purely van der Waals interactions were expected at 3.24 Å, which is the sum of the vdW radii of Ag (1.72 Å) and O (1.52 Å), while covalent radii are 1.45 (Ag) and 0.66 Å (O), which would provide Ag-O distances close to 2.1 Å for covalent bonding. Similar considerations apply, mutatis mutandis, for the Ag-N bond distances. We can describe this situation as Ag^+^ trapped inside a {N_2_O_4_} cage. It is interesting to compare the observed bonding arrangement around Ag^+^ with other reported data. We used the AgO_4_N_2_ coordination environment as the criterion for the structural search in CCDC. The relevant entries are summarized in Table 1. All hits were divided into five groups based on the ligand nature. The first one included complexes with sterically fixed pincer-type ligands based on pyridine as a central fragment and two O-donor arms. In such complexes, the Ag-N bonds were significantly shorter than Ag-O bonds.

However, in the case of HATSAZ, two tridentate ligands completed the coordination sphere of Ag^+^. Therefore, the discrimination between Ag-O and Ag-N bond distances cannot be done properly.

The second group combined complexes with crown-ethers and cryptands. In this environment, Ag^+^ was fixed inside the cages wherein Ag-O contacts were largely determined by ligand geometry. The Ag-N bond distances were shorter than Ag-O, except for TIBQIE, wherein the coordination sphere of Ag^+^ may include additional Ag-π donation (c.a. at 2.83 Å) and Ag-F contacts (2.71 Å). In this case, the bonding nature should be further studied through quantum chemical calculations, to shed light on the Ag^+^ complexes with very high coordination numbers. Remarkably, for such a group of complexes, coordination numbers were lower than the number of donor atoms.

The third group was constituted by POM-based complexes. In this group, an uncommon Ag^+^ coordination by the POM was observed in the case of ZULXOV, wherein there were two types of Ag (namely, Ag1 and Ag2), with a significant Ag-X bond distance distribution. Curiously, the values of d(Ag1-O) ranging from 1.960 to 2.097 Å were too short for Ag and might be attributed to the presence of a 3d metal.

Within the fourth group, no anomalies in the Ag^+^ coordination were revealed. The comparison of the bonding distribution around Ag followed the same trends as in the previous groups.

In the fifth group, more structures featured anomalous geometries. In most cases, the atom assigned as Ag looks more like a 3d metal, rather than Ag. In the case of ELEGIM structure, the “Ag” atom had CN 9 and might be a lanthanide. 

From this structural analysis, it can be seen that the significant difference between Ag-N (shorter) and Ag-O (longer) bond distances is a rule. There are only two cases wherein this rule is infringed, *viz.* in SENJIE and TIBQIE. This might be due to a wrong refinement of N and O atoms or impurities in the initial ligands. High coordination numbers can be achieved only with the use of pincer ligands with fixed geometry, and there is no possibility to reach CN 6 (4 O and 2 N donor atoms) for Ag^+^ using mono or bidentate ligands. The example reported in this work, therefore, represents a unique case in the literature, which reveals a hitherto hidden aspect of Ag^+^ coordination behavior.

### 2.2. Quantum Chemical Calculations

The natural bond orbital (NBO) analysis was conducted on the electron density of the optimized cluster to clarify the nature of the Ag-involving electron density delocalization interactions (ESI, Appendix A). Ag may play the role both of electron density donor and acceptor. As a donor, the NBO of Ag involved are core orbitals (CR 100% *p* in nature; with occupancy > 1.998) or lone pairs (LP 100% *d* in nature; with occupancy > 1.990). From Ag, the electron density shifted to empty virtual Rydberg orbitals (RY*) of C atoms or bond orbital BD*(C-C). When Ag plays the role of acceptor, the virtual NBO of Ag involved are 4 LP* and 6 RY*. Their nature and occupancy is variegated; however, 2 LP* are almost 100% p orbitals; 1 LP* is s^33%^p^67%^; 1 LP* is s^65%^p^35%^; RY* orbitals involve d orbitals (in particular, one RY* is 100% d). In these cases, the donors are CR belonging to O, N, C, or Mo atoms, BD(C-C), BD(C-N), or BD(C-H).

The non-covalent interaction (NCI) analysis confirmed that Ag has CN = 6, showing four strongly attractive interactions (Figure 4, blue regions) with as many O atoms (red arrows) + 2 with as many N atoms (blue arrows), involving each Ag atom. In those regions, bond critical points (BCPs) were identified by means of the topological analysis based on the atoms-in-molecules (AIM) theory (ESI, Appendix A).

The UV–Vis absorption spectrum (Figure 5a) indicates two main bands, which do not represent two single transitions, but cover multiple transitions. The four transitions with higher intensities are found at 478 and 303 nm, each of which shows double degeneration. They are characterized by a complex composition in terms of molecular orbitals, and we hence decided to analyze their nature by means of natural transition orbitals (NTO) analysis (Figure 5b). Both absorption bands corresponded to CT transitions, and in both cases, Ag d orbitals donated electron density. The band at 478 nm was mostly Ag → L (the [Mo_8_O_26_]^4−^ ligand was almost excluded from this electronic transition), and is dipolar in nature. Contrary to this, the other band at 303 nm involved the octamolybdate and was quadrupolar in nature.

### 2.3. Electrochemistry

The structure of **1** combines a redox-active polymolybdate, Ag^+^, and non-innocent dpp-bian ligand. Since the [{Ag(L)_x_}_2_Mo_8_O_26_]^2−^ complexes are unstable in solution [46], their electrochemical behavior can be correctly studied only in solid state.

The redox behavior of **1** in the paste electrode in acetonitrile was studied with cyclic voltammetry (Figure 6, black curve). For the sake of comparison, a cyclic voltammogram for the (Bu_4_N)_4_[Mo_8_O_26_] under the same conditions was recorded (Figure 6, red curve). The comparison of paste electrodes behavior and blank solution is presented in Appendix A (Appendix A). The corresponding data are provided in Table 2 and Table 3. In both cases, a large number of quasi-reversible and irreversible electrochemical transitions were observed, *viz.* from six (for the precursor) to eight (for the complex). The redox properties of the precursor were due to its ability to undergo multistage reduction processes, which is a characteristic feature of POMs. The presence of an additional redox-active dpp-bian fragment in the complex, known for its ability to multielectron reduction, increased the number of electrochemical transitions.

The electrochemistry of octamolybdates was reported in a number of works. Xie et al. reported synthesis and electrochemical behavior of [MoO_3_(TPMA)]·4H_2_O, {[Cu(TPMA)(H_2_O)]_2_(β-Mo_8_O_26_)·3H_2_O}_4_, [Co(TPMA)(β-Mo_8_O_26_)_0.5_], [Co(TPMA)(γ-Mo_8_O_26_)_0.5_], and [Cu_3_(TPMA)_2_(1,3-ttb)_2_(β-Mo_8_O_26_)]·H_2_O (TPMA = tris[(2-pyridyl)methyl]amine; 1,3-ttb = 1-(tetrazo-5-yl)-3-(triazo-1-yl)benzene) [49]. The electrochemical properties were measured for paste electrodes (**CPE**) in 0.5 M Na_2_SO_4_ + 0.1 M H_2_SO_4_ aqueous solution at different scan rates. The authors postulate that “when the scan rate is 100 mV·s^−1^, the mean peak potentials E_1/2_ = (E_p_a + E_p_c)/_2_ are +185 mV (I–I′) for [MoO_3_(TPMA)]·4H_2_O-CPE, 367 mV (I–I′) and 204 mV (II–II′) for [Co(TPMA)(β-Mo_8_O_26_)_0.5_]-**CPE**, 228 mV (I–I′) and 120 mV (II–II′) for [Co(TPMA)(γ-Mo_8_O_26_)_0.5_]-**CPE**, and 204 mV (I–I′) and 83 mV (II–II′) for [Cu_3_(TPMA)_2_(1,3-ttb)_2_(β-Mo_8_O_26_)]·H_2_O-**CPE**′′. They assigned these processes to Mo^6+^ reduction. However, in our opinion, ligand-centered redox reactions or redox reactions involving a heterometal that might occur in this region cannot be excluded. Indeed, for the initial (Bu_4_N)_4_[Mo_8_O_26_], we did not observe any reduction peaks in the positive region. Only oxidation peaks (peaks 4, 5, and 6 in the red curve in Figure 6) were found in the anodic curve and they can be attributed to the oxidation of electrochemically generated reduced species during cycling. On the other hand, the absence of cathodic peaks in the positive region for (Bu_4_N)_4_[Mo_8_O_26_] in comparison with related complexes can be explained by the influence of the medium and the shift of the Mo^6+^ reduction peaks to a more anodic region caused by the participation of protons in redox processes.

The same redox behavior was reported by Hoi et al. for H_2_{[Cu_3_(PCAP)_4_(H_2_O)_2_](β-Mo_8_O_26_)}·10H_2_O, H{[Cu(PCAP)(H_2_O)](β-Mo_8_O_26_)_0.5_} and H_2_[Co(H_2_O)_6_][Co_2_(PCAP)_4_(γ-Mo_8_O_26_)(H_2_O)_2_]·10H_2_O [50]. In both papers, the authors reported the electrochemical activity of the Cu-Mo_8_O_26_ complexes toward H_2_O_2_ reduction.

On the other hand, solid-state electrochemistry of bulk molybdates has been attracting increasing interest for possible technological applications, such as MgMoO_4_, which has been used for supercapacitor and battery applications [51], or CoMoO_4_, which has shown a specific capacitance of 79 F·g^−1^, an energy density of 21 W h·kg^−1^, and excellent cycling stability (retaining over 75% of its initial capacitance after 2000 cycles), which makes it a very promising candidate for large-scale energy-storage applications.

Recently, Schöfberger et al. reported the catalytic activity of [Ag^I^(bian)_2_]BF_4_ in electrochemical CO_2_ reduction [52]. A drop-casted ink of [Ag^I^(4-OMe-Ph-bian)_2_]BF_4_ immobilized onto carbon paper gas diffusion electrodes in a flow cell with 1.0 M KHCO_3_ aqueous electrolyte, resulting in Faradic efficiency of 51% for CO at a current density of 50 mA cm^−2^. This observation prompted us to use (Bu_4_N)_2_[β-{Ag(dpp-bian)}_2_Mo_8_O_26_] as the electroactive matter of the carbon-based paste electrode for CO_2_ reduction. The electrochemical activity was tested in a CO_2_-saturated 1.0 M KHCO_3_ aqueous solution. Figure 7 shows a CV series recorded in the range of −1.4/1.4 V relative to a saturated Ag/AgCl reference electrode at a 10 mV/s scan rate for a paste electrode containing **1**. The red arrow in Figure 7 shows the cathodic potential sweep direction; the beginning of the first cycle corresponds to the 0.50 V point. In the region from approx. −0.9 to −1.2 V, there is a cathodic plateau that is most likely related to CO_2_ reduction. The black arrow on this plateau displays the sequential increase in current from the first to subsequent scans. At potentials below −1.2 and above 1.0 V, there was a sharp increase in current due to HER and OER, respectively. The voltammograms also revealed regions at potentials of ~0.10 (cathodic current peaks) and ~0.50 V (anodic current peaks), which are responsible for the presence of complex **1** in the paste. The dependence of the current in the peak on the square root of the potential scan (Appendix A in Appendix A) indicates the diffusion nature of the current for the redox reaction of complex **1** and its stability.

Comparison of CVs (Figure 8) recorded in the absence (red curve, blank) and in the presence of complex **1** (grey curve) in a 1.0 M KHCO_3_ solution saturated with CO_2_ or in a solution unsaturated with CO_2_ (blue curve) showed that, in the region of cathodic currents, the area (on the gray curve) appears with potentials in the region of−1.16/−0.87 V (relative to Ag/AgCl), which, apparently, is responsible for the electrocatalysis of CO_2_ reduction to less deeply oxidized products. The conversion to the RHE scale provided the potential ranging from −1.35 to −1.06 V. According to the literature data, the direct reduction involved the following first step:CO_2_ + e = CO_2_^•−^E° = −1.48 V (vs. RHE)(1)

When CO_2_ was saturated in the presence of **1**, an increase in the cathodic current was observed (Figure 8). In this case, the difference in an area between the blue (**1**) and gray (**1** + CO_2_) cathodic currents reached 5.0 mC or 29 mC/cm^2^ in the range of −1.16/−0.87 V. The total difference between the red (blank + CO_2_) and gray currents was 34 mC or 176 mC/cm^2^.

The cathodic peak at approximately −0.5 V observed on the red curve (Figure 8) refers to the Nafion matrix containing acidic SO_3_H groups capable of ion exchange [53]. This peak is slightly shifted towards positive potentials for the paste containing **1** (up to −0.25 V). On the other hand, in the potential region of 0.1–0.2 and 0.5–0.6 V (gray and blue curves), cathodic and anodic peaks were observed, marking the presence of **1** in the pastes.

Typically, currents observed under anhydrous conditions and in the solvent–water mixtures differ due to the presence of protons accessing alternative reduction mechanisms. It is well known that, without external proton sources, electrocatalytic CO_2_ reduction can proceed: (i) as reductive disproportionation forming carbonate and CO, or (ii) as dimerization of CO_2_^•−^ anion-radical to oxalate, while proton addition opens pathways to numerous reduction products, such as CO, formic acid, methanol, methane, and higher carbon products, e.g., ethylene, ethanol, and acetic acid [54,55,56,57,58,59]. The possible reaction mechanism and catalytic activity of Ag^+^ coordinated by different BIANs are currently the object of a dedicated investigation in our group, both from an experimental and computational point of view. However, in the specific case investigated in the present work, taking into account the activity of the complex in the solid state, the through-space electron transfer from the reduced complex to CO_2_ is the only possible explanation. In fact, the direct coordination of CO_2_ cannot be hypothesized due to lack of coordination space near to Ag^+^ caused by the bulky side groups installed onto BIAN molecules.

To check this hypothesis, we calculated the first reduced state of [β-{Ag(dpp-bian)}_2_Mo_8_O_26_]. The obtained spin density profile is shown in Figure 9.

According to the calculated data, for the first reduced state (induced during electroreduction) the spin density was mostly located over two BIAN ligands. Moreover, its small portion affects both Ag atoms. Thus, we can conclude that electron transfer to CO_2_ can be indeed realized from the combination of Ag and BIAN electronic states.

The formation of formic acid as a possible product of CO_2_ reduction could be judged from a comparison of CV data for pure HCOOH in 0.1 M Na_2_SO_4_ and the CV data for a paste electrode containing **1** in the presence of CO_2_ under the same conditions (Figure 10).

On the CV of the blank paste (blue curves in Figure 10) in the range of −1.40/−1.00 V, a region of cathodic reduction of formic acid, was observed. This region intersected with an interval of −1.25/−0.80 V, which is responsible for the reduction of CO_2_ in the paste containing **1** (orange curves). Note that the other characteristic regions on the orange curves in Figure 10 (Na_2_SO_4_) are close to the data where the supporting electrolyte was KHCO_3_ (Figure 8): Nafion matrix peak at−0.5/−0.4 V; cathodic and anodic peaks of **1** at 0.1/0.2 and 0.3/0.5 V.

## 3. Materials and Methods

### 3.1. General Information

(NBu_4_)_4_[β-Mo_8_O_26_] and dpp-bian [60] were synthesized according to the standard procedures. All other reagents were of commercial purity (Sigma Aldrich, Buchs, Switzerland) and were used without any purification. Organic solvents (DMF and diethyl ether ((C_2_H_5_)_2_O)) were dried by standard methods before use. Elemental analysis was carried out on a Eurovector EA 3000 CHN analyzer. IR spectrum was recorded in the 4000–300 cm^−1^ range with a Perkin-Elmer System 2000 FTIR spectrometer (KBr pellets). DR was carried out on a UV–Vis–NIR spectrophotometer UV–3101 PC in an integrating sphere in a range of 240–800 nm (gap width 5 nm) at room temperature.

### 3.2. Synthesis of (Bu_4_N)_2_[{Ag(dpp-bian)}_2_Mo_8_O_26_] (**1**)

A mixture of (Bu_4_N)_4_Mo_8_O_26_ (100 mg, 46 mmol) and AgCF_3_SO_3_ (23.7 mg 92 mmol) was dissolved in DMF (2 mL). The solution was stirred for 30 min at 60 °C. After the reagents were dissolved, dpp-bian (46.2 mg, 92 mmol) was added. The mixture was stirred for 30 min at room temperature. The resulting orange solid was filtered, washed with diethyl ether, and dried in a vacuum. Yield 102 mg (76%). Crystals of **1**, suitable for X-ray diffraction analysis, were obtained by diffusion of diethyl ether onto the reaction solution of **1** in DMF. Anal. Calc. for C_110_H_152_N_6_O_26_Mo_8_Ag_2_: C 43.29; 5.31; N 2.91%. Found: C 43.1, H 5.1, N 2.8% IR (Appendix A, KBr, cm^−1^): 3452 (m), 3064 (w), 2960 (vs), 2929 (s), 2872 (m), 1869 (w), 1662 (m), 1626 (m), 1587 (m), 1518 (w), 1483 (m), 1464 (m), 1431 (m), 1383 (w), 1362 (w), 1327 (w), 1279 (w), 1254 (w), 1223 (w), 1192 (w), 1150 (w), 1107 (w), 1061 (w), 1045 (w), 943 (vs), 926 (vs), 905 (vs), 839 (s), 808 (w), 795 (w), 781 (m), 756 (m), 727 (m), 704 (s), 656 (m), 573 (w), 556 (w), 523 (m), 480 (w), 465 (w), 448 (w), 413 (m). TGA (Appendix A).

### 3.3. SCXRD

Single-crystal XRD data for **1** were collected with a Bruker D8 Venture diffractometer (Bruker, Billerica, MA, USA) with a CMOS PHOTON III detector and IµS 3.0 source (Mo Kα radiation, λ = 0.71073 Å, Montel mirror optics, nitrogen flow thermostat). φ- and ω-scan techniques were employed to collect data for crystal structure refinement; full φ-scans were used for investigation of temperature dependence of unit cell parameters in 80–280 K range (increasing of temperature by 5–10 K steps was used). The unit cell parameters were refined using APEX3 Determine Unit Cell routines with the refinement of beam position instrumental parameters and may have similar systematic errors (~0.01 Å, ~0.1°) [61]. Integration and absorption corrections were applied with the use of APEX3 program suite (Bruker APEX3 software suite (APEX3 v.2019.1-0, SADABS v.2016/2, SAINT v.8.40a), Bruker Nonius (2003–2004), Bruker AXS (2005–2018), Bruker Nano (2019): Madison, WI, USA). Structures were solved by SHELXT [62] and refined by full-matrix least-squares treatment against |F|^2^ in anisotropic approximation with SHELX 2014/7 [63] in the ShelXle program [64]. The main geometrical parameters are present in Appendix A. H-atoms were refined in calculated positions. One of TBA^+^ was disordered over two closed positions with 0.4/0.6 occupancies. C-atoms of such parts were refined isotropically.

The crystallographic data were deposited In the Cambridge Crystallographic Data Centre under the deposition codes, CCDC 2189249.

### 3.4. Quantum Chemical Calculations

The molecular geometry of the monomer and model dimer was fully optimized using density functional theory (DFT), as well as using the long-range corrected hybrid functional CAM-B3LYP [65] coupled with the Pople 6-311+G** triple-ζ basis set for H, C, N, and O [66,67,68,69,70,71,72,73,74], the LANL2DZ double-ζ basis set on the Mo and Ag valence shell, and the LANL2DZ effective core potential on the core [75,76,77,78]. The vibrational frequencies and thermochemical values were subsequently computed at the same levels of theory, within the harmonic approximation, at T = 298.15 K and p = 1 atm; no imaginary frequencies were found. 

The natural bond orbital (NBO) [79,80,81,82,83,84,85], the atoms-in-molecules (AIM) Bader’s theory [86,87,88], and the non-covalent interaction [89,90,91,92] analyses were carried out on the ground state electron densities to investigate the nature of the interactions.

The UV–Vis absorption spectra for the equilibrium geometries were calculated at a time-dependent density functional theory (TD-DFT) level, accounting for S_0_ → S_n_ (*n* = 1 to 120) transitions. The nature of the vertical excited electronic state was analyzed via natural transition orbital (NTO) analysis [93].

The integration grid was set to 250 radial shells and 974 angular points. The convergence criteria for the self-consistent field were set to 10^−12^ for the RMS change in the density matrix and 10^−10^ for the maximum change in the density matrix. The convergence criteria for optimizations were set to 2 × 10^−6^ a.u. for the maximum force, 1 × 10^−6^ a.u. for the RMS force, 6 × 10^−6^ a.u. for the maximum displacement, and 4 × 10^−6^ a.u. for the RMS displacement.

The location of the CPs and subsequent calculation of SF values were performed using a modified version of the PROAIMV program [94]. The NCI analyses were conducted and plotted using homemade code. All the other calculations were performed using the GAUSSIAN G16.A03 package [95].

### 3.5. Electrochemistry

The CV measurements were carried out in a three-electrode cell with a paste electrode, an auxiliary electrode, and a reference electrode using a potentiostat-galvanostat P-45X (Elins, Chernogolovka, Russia) at 25 °C. The description of the cell and the measurement procedure have been previously reported [96,97,98]. The composition of the paste included carbon powder:EAS:Nafion = 100:10:4 mg (Nafion, 10% aqueous dispersion calculated on dry matter) for the complex **1** and the precursor. Based on the weights of the dry mixture, the concentration of **1** in the paste was ~0.030 and the precursor ~0.070 mol/kg. Initially, the measurements were carried out in the acetonitrile solutions containing 0.10 M Bu_4_NPF_6_ (supporting electrolyte). In total, in these experiments, three types of pastes were tested, viz. the pastes contained the complex **1**, the precursor, and the blank (without EAS). The blank paste contained only the carbon powder and the Nafion dispersion in the ratio 100:4 mg. 

CO_2_ reduction experiments were performed in an aqueous 1.0 M KHCO_3_ solution. Based on the weight of the dry mixture, the concentration of **1** in the paste was 0.030 mol/kg. The blank paste contained only the carbon powder and the Nafion dispersion (as in experiments with acetonitrile solutions). First, the electrochemical experiments were carried out in solutions saturated with CO_2_. Then, the measurements were continued in the same solution, but without its saturation with CO_2_. Finally, to compare the obtained results, measurements were also carried out on the blank pastes.

Some additional experiments were done with paste electrodes in an aqueous solution of 0.10 M Na_2_SO_4_. The composition of the paste was the same as in the previous series. In this case, we tried to find the difference in the CV behavior between the blank paste in solution with the addition of formic acid (0.038 M) and the paste after saturation with CO_2_.

## 4. Conclusions

Three components, and five building blocks’ self-assembly, produced a [{Ag(dpp-bian)}_2_Mo_8_O_26_]^2−^ complex which had a non-typical coordination environment for Ag^+^ cation, wherein Ag had six comparable distances to both O and N donor atoms. There was no remarkable difference between all Ag-N and Ag-O bond lengths; however, the Ag-N bond was revealed to be covalent in nature, whereas Ag-O had a predominant ionic character. Quantum–chemical calculations confirmed the CN of Ag^+^ cation and indicated the charge-transfer character of the two bands in the UV–Vis absorption spectrum. Solid (Bu_4_N)_2_[β-{Ag(dpp-bian)}_2_Mo_8_O_26_] showed rich electrochemistry which involved both organic components and octamolybdate ligands with an electrochemical window of ca. 2.5 V between two quasi-reversible processes in both extremes of the cycling route. The complex also demonstrated a catalytic activity toward electrochemical CO_2_ reduction. CO_2_ activation most likely occurs at the Ag-bian redox site.

## Figures and Tables

**Figure 1 molecules-27-06961-f001:**
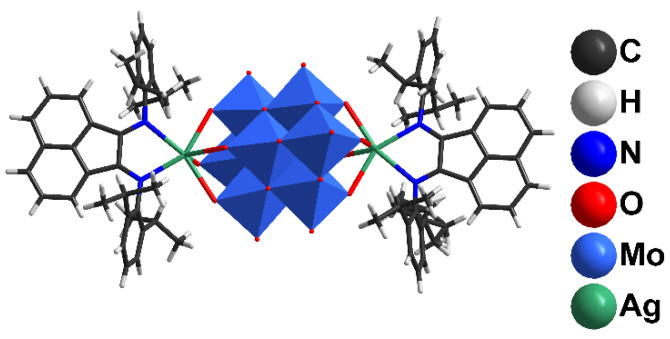
The structure of [β-{Ag(dpp-bian)}_2_Mo_8_O_26_]^2−^.

**Figure 2 molecules-27-06961-f002:**
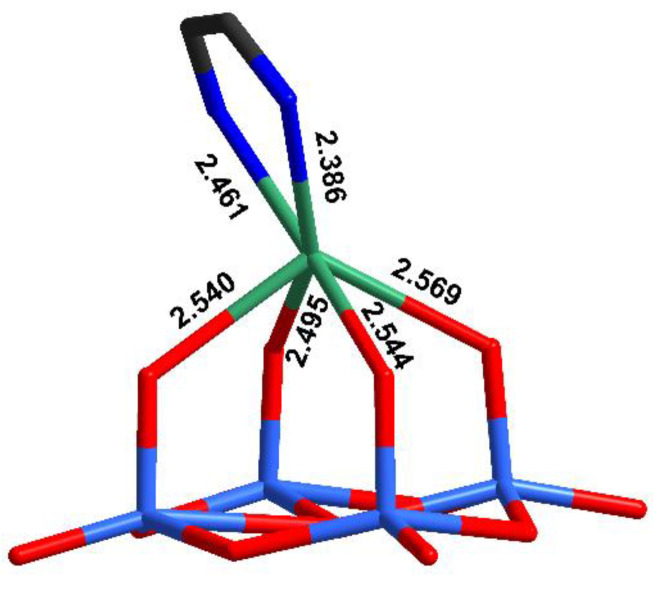
The coordination environment of Ag^+^ in the crystal structure of **1**.

**Figure 3 molecules-27-06961-f003:**
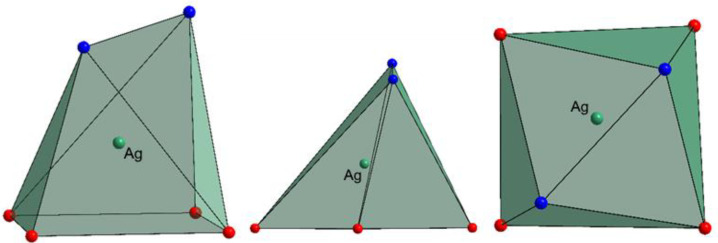
The coordination polyhedron of Ag+. O-atoms are red, N-atoms are blue.

**Figure 4 molecules-27-06961-f004:**
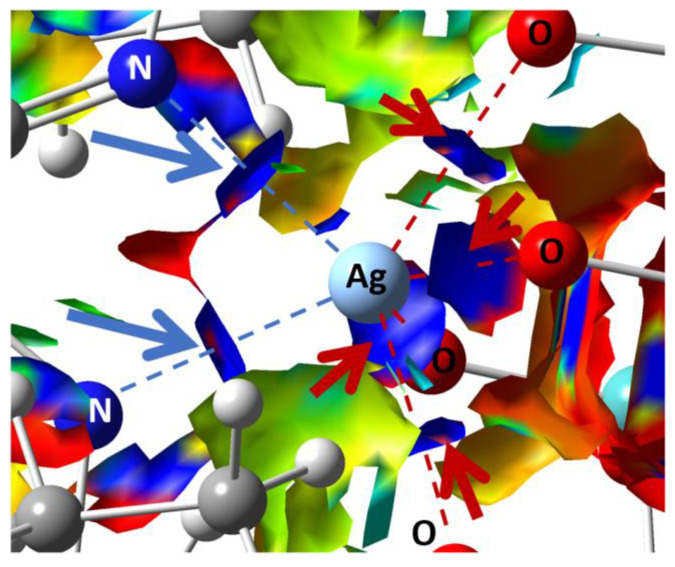
Plots of the isosurfaces of non-covalent interaction index mapped over the reduced density gradient (*s* = 1/2). Legend of colors: −0.012 a.u. (red; repulsion) to +0.012 a.u. (blue; attraction). Ag-O and Ag-N attractive NCIs are pointed out by red and blue arrows, respectively. The calculations were conducted on the optimized geometry.

**Figure 5 molecules-27-06961-f005:**
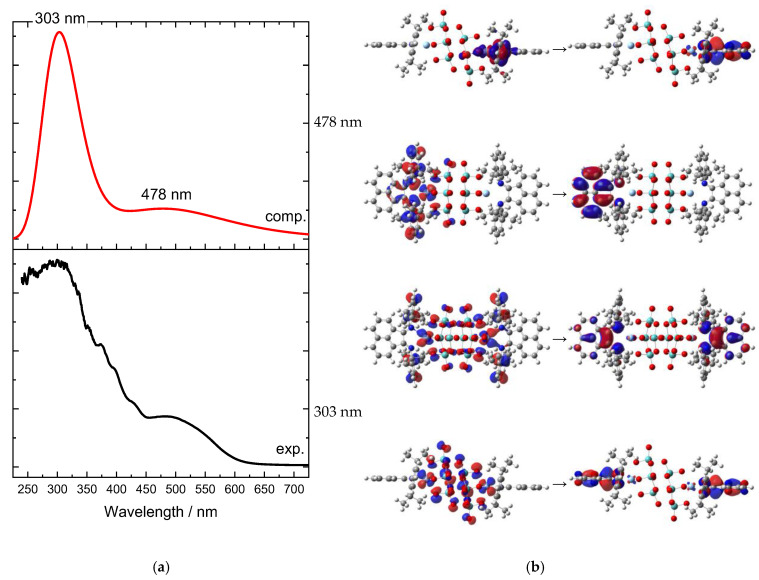
(**a**) Experimental (black line) and computed UV–Vis absorption spectra (red line) of complex **1** in the wavelengths’ region from 250 to 700 nm. No scale factor applied. (**b**) Natural transition orbital (NTO) analysis of the main transitions corresponding to the two electronic bands; two perspectives are depicted for each transition. Calculations were performed on the optimized geometry.

**Figure 6 molecules-27-06961-f006:**
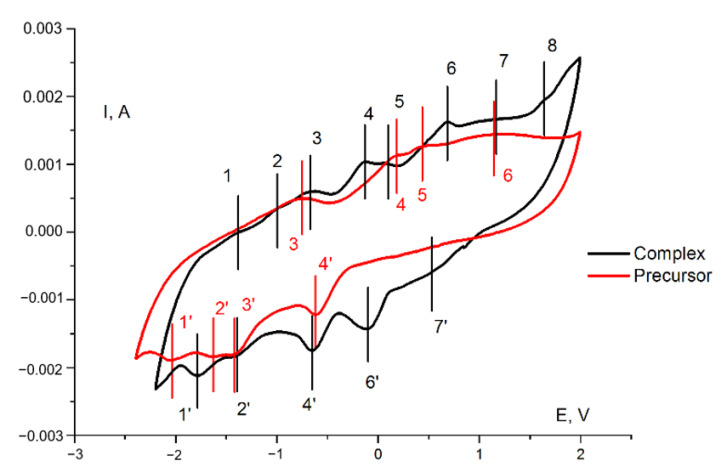
Cyclic voltammogram (second cycle) of **1** (black curve) and a (Bu_4_N)_4_[Mo_8_O_26_] (red curve) in a paste electrode in acetonitrile with supporting electrolyte (0.10 M Bu_4_NPF_6_) at a sweep rate of 20 mV/s. Potentials vs. saturated Ag/AgCl reference electrode.

**Figure 7 molecules-27-06961-f007:**
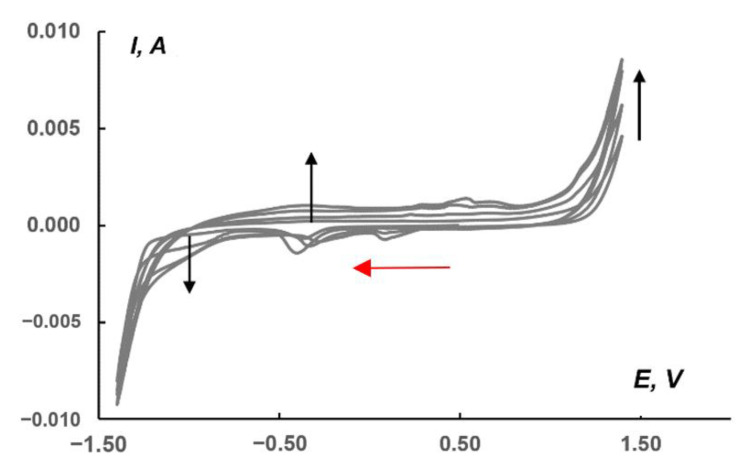
CVs (four consecutive cycles) for the paste electrode including carbon powder, **1**, and Nafion 10% aqueous dispersion in the ratio of 100:10:4 (the last component calculated as the dry mixture). The measurements were made at 10 mV/s scan rate in CO_2_-saturated 1.0 M KHCO_3_. Black arrows indicate the sequence of cycles, and red ones—the starting point and direction of CV starting potential. Potentials vs. saturated Ag/AgCl reference electrode.

**Figure 8 molecules-27-06961-f008:**
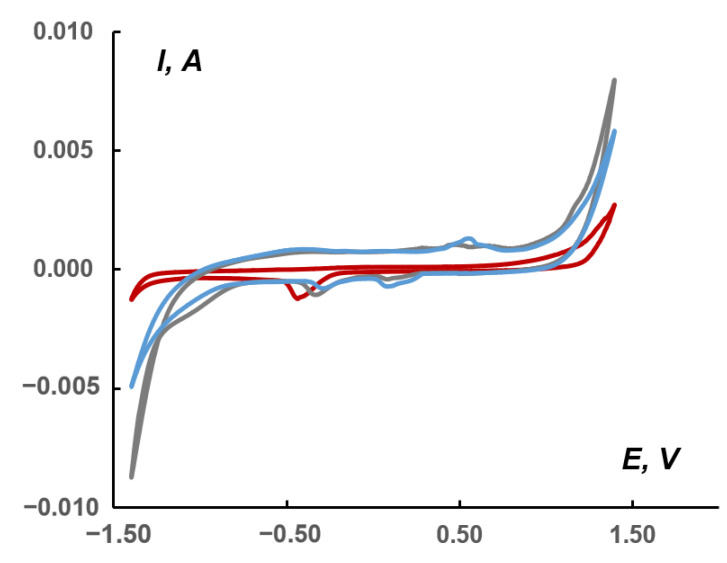
CVs (second cycle) for the following pastes: red curve—blank in the CO_2_ saturated solution (without electroactive compound); blue—containing **1**; and gray—containing **1** in the CO_2_ saturated solution; 1.0 M KHCO_3_ aqueous solutions at 10 mV/s scan rate. Potentials vs. saturated Ag/AgCl reference electrode.

**Figure 9 molecules-27-06961-f009:**
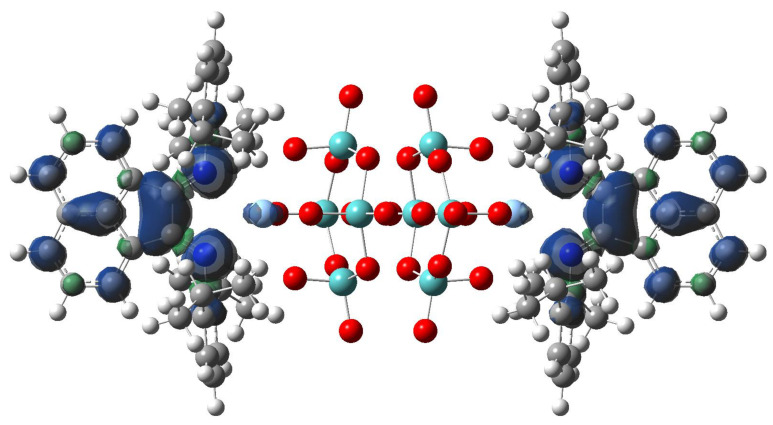
The spin density distribution on [{Ag(dpp-bian)}_2_Mo_8_O_26_]^3•−^ anion.

**Figure 10 molecules-27-06961-f010:**
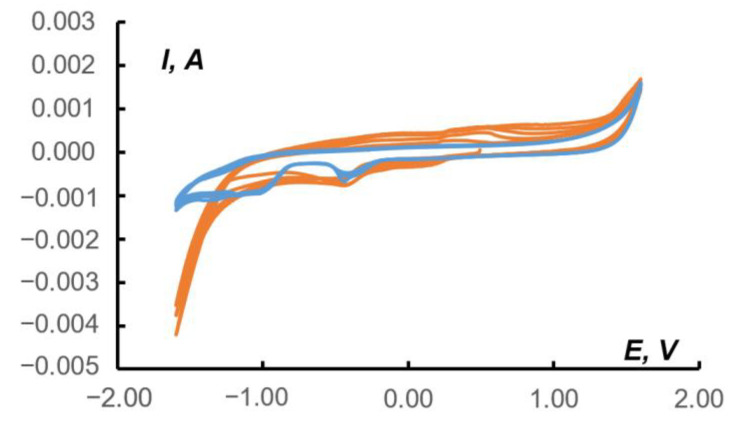
The CV data (first five cycles) for 0.0373 M HCOOH in 0.1 M Na_2_SO_4_ aqueous solution (blue curves); CV data (first five cycles) for the paste electrode containing **1** after CO_2_ exposure into the 0.1 M Na_2_SO_4_ aqueous solution (orange curves). Potentials vs. saturated Ag/AgCl reference electrode. Scan rate 10 mV/s.

**Table 1 molecules-27-06961-t001:** The analysis of CCDC data for Ag complexes with two N and four O-atoms in the coordination sphere.

REFCODE	d(Ag-N)/Å	d(Ag-O)/Å	CN
**Group 1—Fixed Geometry Ligands**			
FOWWIA	2.316, 2.355	2.532–2.641	6
GIFRIZ	2.362	2.502–2.653	6
HATSAZ	2.470	2.420, 2.478	6
IXEFAU	2.325	2.491–2.573	6
IXEFEY	2.326	2.478. 2.613	6
QOMQER	2.324	2.558, 2.617	6
VAZDEF	2.315	2.522, 2.588	6
**Group 2—Crown-Ethers and Cryptands**			
FAFQUY	2.483	2.923	4
GOXTAO	2.275, 2.304	2.681–2.745	2
KISPUZ	2.437	2.558–2.748	4
PIMCET	2.219, 2.235	2.736, 2.850	2
QIFPEA	2.323, 2.342	2.564–2.741	3
QIFPIE	2.392	2.562–2.737	3
ROQLOY	2.372	2.834	2
SENJIE	2.208, 2.721	2.498–2.557	5
TIBQIE	2.585, 2.694	2.478–2.626	5
YUHDIO	2.346, 2.395	2.608–2.684	4
**Group 3—POM Complexes**			
PUCYES	2.262, 2.349	2.463, 2.606, 2.678	3
ODEQOE	2.389	2.564–2.632	6
XUJSAX	2.270, 2.360	2.385–2.860	3
ZULXOV (Ag2)	2.340, 2.359	2.530–2.643	6
**Group 4—Other Complexes**			
PICVII	2.303, 2.320	2.529–2.864	6
PURZUY	2.232, 2.283	2.547–2.777	3
CAVWAY	2.249	2.657	4
FEJFEF	2.273, 2.301	2.517–2.574	5
HAXFOB	2.358	2.580–2.619	6
KIJRIG	2.400	2.456–2.633	6
LADFED	2.338, 2340	2.291–2.736	6
MEJBAE	2.270	2.548–2.571	6
MIZAGN	2.372	2.600–2.628	6
NAKLOB	2.382	2.605–2.619	6
NELMAT	2.415	2.446–2.790	4
RAWVER	2.332	2.609–2.625	6
YIXDAM	2.402	2.508–2.558	6
**Group 5—Questionable Structures**			
ZULXOV (Ag1)	1.960–2.097	2.586, 2.311	6
WOKRIY (3d)	2.176, 2.181	2.867, 2.931	4
SIKJIH (3d)	2.141, 2.180	2.150, 2.108	6
ELEGIM (Ln)	2.440, 2.544	2.470–2.569	9
PYRCAG (3d)	2.086, 2.210	2.203–2.542	6
SEHSAA (Ag2)	2.159	3.010–3.074	2
SEHSAA (Ag1)	2.161	2.868–3.178	2

**Table 2 molecules-27-06961-t002:** Characteristics of half-wave potentials, as well as anodic and cathodic current peaks of the complex (Bu_4_N)_2_[β-{Ag(dpp-bian)}_2_Mo_8_O_26_]. Potentials vs. saturated Ag/AgCl reference electrode.

E_1/2_ (**1,1′**), V	−1.585	ΔE, V	0.400	quasi-reversible
E^a^, V	−1.390	I^a^, A	−1.7 × 10^−5^	
E^c^, V	−1.790	I^c^, A	−2.1 × 10^−3^	
E_1/2_ (**2,2′**), V	−1.200	ΔE, V	0.400	quasi-reversible
E^a^, V	−1.000	I^a^, A	3.4 × 10^−4^	
E^c^, V	−1.400	I^c^, A	−1.8 × 10^−3^	
E_1/2_ (**3,3′**), V		ΔE, V		irreversible
E^a^, V	−0.670	I^a^, A	5.9 × 10^−4^	
E^c^, V		I^c^, A		
E_1/2_ (**4,4′**), V	−0.390	ΔE, V	0.520	quasi-reversible
E^a^, V	−0.130	I^a^, A	1.0 × 10^−3^	
E^c^, V	−0.650	I^c^, A	−1.8 × 10^−3^	
E_1/2_ (**5,5′**), V		ΔE, V		irreversible
E^a^, V	0.090	I^a^, A	1.0 × 10^−3^	
E^c^, V		I^c^, A		
E_1/2_ (**6,6′**), V	0.290	ΔE, V	0.780	quasi-reversible
E^a^, V	0.680	I^a^, A	1.6 × 10^−3^	
E^c^, V	−0.100	I^c^, A	−1.4 × 10^−3^	
E_1/2_ (**7,7′**), V	0.850	ΔE, V	0.640	quasi-reversible
E^a^, V	1.170	I^a^, A	1.7 × 10^−3^	
E^c^, V	0.530	I^c^, A	6.1 × 10^−4^	
E_1/2_ (**8,8′**), V		ΔE, V		irreversible
E^a^, V	1.650	I^a^, A	2.0 × 10^−3^	
E^c^, V		I^c^, A		

^a^ anodic; ^c^ cathodic

**Table 3 molecules-27-06961-t003:** Characteristics of half-wave potentials, as well as anodic and cathodic current peaks of the (Bu_4_N)_4_[Mo_8_O_26_] precursor.

E_1/2_ (**1,1′**), V		ΔE, V		irreversible
E^a^, V		I^a^, A		
E^c^, V	−2.050	I^c^, A	−1.9 × 10^−3^	
E_1/2_ (**2,2′**), V		ΔE, V		irreversible
E^a^, V		I^a^, A		
E^c^, V	−1.62	I^c^, A	−1.8 × 10^−3^	
E_1/2_ (**3,3′**), V	−1.085	ΔE, V	0.670	quasi-reversible
E^a^, V	−0.750	I^a^, A	5.1 × 10^−4^	
E^c^, V	−1.420	I^c^, A	−1.8 × 10^−3^	
E_1/2_ (**4,4′**), V	−0.218	ΔE, V	0.785	quasi-reversible
E^a^, V	0.175	I^a^, A	1.1 × 10^−3^	
E^c^, V	−0.610	I^c^, A	−1.2 × 10^−3^	
E_1/2_ (**5,5′**), V		ΔE, V		irreversible
E^a^, V	0.430	I^a^, A	1.3 × 10^−3^	
E^c^, V		I^c^, A		
E_1/2_ (**6,6′**), V		ΔE, V		irreversible
E^a^, V	1.140	I^a^, A	1.45 × 10^−3^	
E^c^, V		I^c^, A		

^a^ anodic; ^c^ cathodic

## Data Availability

The crystallographic data have been deposited in the Cambridge Crystallographic Data Centre under the deposition codes CCDC 2189249.

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
