# Peer review of "Trapping of Ag+ into a Perfect Six-Coordinated Environment: Structural Analysis, Quantum Chemical Calculations and Electrochemistry"

_molecules, 2022, doi:10.3390/molecules27206961_

Round 1

Reviewer 1 Report

The present paper by Pavel et al. deals with Ag+ encapsulation within a six-coordination environment. The authors have studied the self-assembled complex in detail using SCXRD, cyclic voltammetry and quantum chemical calculations. I believe that the work presented is suitable for publication in molecules. However, some minor corrections can be included before acceptance.

Minor points

1) Fig.S1 (supporting inf.) is not described in the text. Why does the peak signals (experimental and calculated) appear in opposite direction? If negative intensities, the authors should write in detail.

2) Crystal color or picture of complex 1 should be provided in the text if possible.

3) It’s not clear whose spectra is shown in figure 5. If it is for complex 1, it should be clearly mentioned.

Author Response

Referee 1

1) Fig.S1 (supporting inf.) is not described in the text. Why does the peak signals (experimental and calculated) appear in opposite direction? If negative intensities, the authors should write in detail.

Authors reply: We thank Referee #1 for this comment. The peak signals are located in different directions for easier perception. All calculated signals for complex 1 also have positive intensities.

2) Crystal color or picture of complex 1 should be provided in the text if possible.

Authors reply: The shape and color of the crystals have been added to section 2.1.

3) It’s not clear whose spectra is shown in figure 5. If it is for complex 1, it should be clearly mentioned.

Authors reply: We appreciate this observation. It is spectra of complex 1. The title of the figure 5 has been corrected.

Reviewer 2 Report

The paper reported the synthesis, structure , and catalytic activity of a Ag(I) coordination compound. The whole manuscript is smooth and the result is of interest. I recommend the publication of this article after minor revisions mentioned below.

1.       Please remove any B level alerts in the checkCIF file.

2. Please compare the catalytic activity of this coordination compound with that of reported ones.

Author Response

Referee 2

  1. Please remove any B level alerts in the checkCIF file.

Authors reply: We thank Referee #2 for this comment. It has been fixed. Corresponding changes have been made to the Table S1

  1. Please compare the catalytic activity of this coordination compound with that of reported ones.

Authors reply: At this stage, we cannot carry out a correct comparison of catalytic activity, since we have only preliminary data obtained by the CV method. To do this, it is necessary to conduct a detailed quantitative study, which we have planned to continue this work.

Reviewer 3 Report

The manuscript “Trapping of Ag+ into a Perfect CN 6 Coordination Environment: Structural Analysis, Quantum Chemical Calculations and Electrochemistry” represents a comprehensive investigation of a new (Bu4N)2[β-{Ag(dpp-bian)}2Mo8O26] complex obtained by self-assembly of (Bu4N)4[β-Mo8O26], AgNO3 and 2-bis[(2,6-diisopropylphenyl)-imino]acenaphthene (dpp-bian) in DMF solution. Ag+ /ligand/polyoxometalate hybrid organic-inorganic systems are of particular interest due to their capability to construct versatile topologies. The novelty of this work consists in using the redox-active dpp-bian ligand as a structural motif, which opens wide opportunities for giving new promising features to such architectures. Single crystal X-ray diffraction, X-ray powder diffraction, diffuse reflectance, infrared spectroscopy and elemental analysis were applied for studying the structural peculiarities of the (Bu4N)2[β-{Ag(dpp-bian)}2Mo8O26] complex. The experimental findings were supported by the results of quantum-chemical calculations. The conclusions were made on unusual coordination environment for Ag+, in which all Ag-N and Ag-O bond lengths are virtually equivalent. Moreover, a catalytic activity toward electrochemical reduction of CO2 was demonstrated by the complex under study, which provides for the manuscript an additional significance. The presented results open prospects for constructing and exploring new multifunctional coordination networks prepared with the use of Ag-based supramolecular building blocks and redox-active nitrogen-donor ligands.

The work corresponds to the profile of the journal and is suitable for publication after taking into account for the following issues:

1) The abbreviation CN (coordination number) in the title of the manuscript may mislead readers, given that it is more often used to refer to the cyano group. Moreover CN abbreviation comprises a word “coordination”, which is already used in the title (Coordination Environment). It is recommended to rephrase the title (for instance, “Trapping of Ag+ into a Perfect Six-Coordinated Environment…”).

2) The first paragraph in the Results and discussion section (“This section may be divided…”) should be removed.

3) In the sections 2.2 and 4.4 there is information on carrying out the AIM analysis (see lines 152-154 and 389-390). However, the corresponding results of calculations are absent. This discrepancy must be eliminated.

4) In the abstract, it is indicated that “Quantum chemical calculations were performed to qualify the nature of the Ag-N/ Ag-O interactions..”. In the manuscript there are discussions on that “Ag may play the role both of electron density donor and acceptor.”, however the conclusion on which exactly the nature of the interactions under study is absent. It is recommended to clarify this point and add this information into the Conclusions.

5) Figure 4 is uninformative. It is recommended either prepare it in a higher quality, or move it to the Supporting Information. 

6) In Figure 5 (b), the NTO for 478 nm are shown in different ways: first, the isosurfaces are from the right and then – from the left. It is recommended to flip one of the perspectives in order to the isosurfaces are on the same side.

7) In Figure 9, spin density at two BIAN ligands is shown, while the caption is “the spin density inside the one-electron reduced complex 1”. It is very unusual when one electron is delocalized over two distant from each other fragments of a molecule. Could the authors clarify, whether it is a two-electron reduced complex 1 (by one electron on each BIAN ligand), or it is a truly one-electron reduced complex?

Author Response

Referee 3

1) The abbreviation CN (coordination number) in the title of the manuscript may mislead readers, given that it is more often used to refer to the cyano group. Moreover CN abbreviation comprises a word “coordination”, which is already used in the title (Coordination Environment). It is recommended to rephrase the title (for instance, “Trapping of Ag+ into a Perfect Six-Coordinated Environment…”).

Authors’ reply: We thank Referee #3 for this comment. It has been fixed.

2) The first paragraph in the Results and discussion section (“This section may be divided…”) should be removed.

Authors’ reply: We appreciate this observation. It has been fixed.

3) In the sections 2.2 and 4.4 there is information on carrying out the AIM analysis (see lines 152-154 and 389-390). However, the corresponding results of calculations are absent. This discrepancy must be eliminated.

Authors’ reply: We are grateful to the Referee for this comment. In the submitted version of the ms we omitted to include the topological results. In the revised version these data were added in ESI (Table S6) which is recalled in the main text (section 2.2)

4) In the abstract, it is indicated that “Quantum chemical calculations were performed to qualify the nature of the Ag-N/ Ag-O interactions.”. In the manuscript there are discussions on that “Ag may play the role both of electron density donor and acceptor.”, however the conclusion on which exactly the nature of the interactions under study is absent. It is recommended to clarify this point and add this information into the Conclusions.

Authors’ reply: We thank the Referee for the recommendation. The nature of the Ag-N and Ag-O interactions is now explicitly stated in the Conclusions of the revised version of the ms, as recommended.

5) Figure 4 is uninformative. It is recommended either prepare it in a higher quality, or move it to the Supporting Information.

Authors’ reply: Thanks to this comment, we realise that for some unknown technical issue, the arrows indicating the interactions (as mentioned in the caption) disappeared, making the figure unreadable. In the revised version of the ms, arrows, interaction lines and atoms’ symbols were added. We apologise for the inconvenience.

6) In Figure 5 (b), the NTO for 478 nm are shown in different ways: first, the isosurfaces are from the right and then – from the left. It is recommended to flip one of the perspectives in order to the isosurfaces are on the same side.

Authors’ reply: We agree with the Referee. In the revised ms, the figure has been changed according to the recommendation.

7) In Figure 9, spin density at two BIAN ligands is shown, while the caption is “the spin density inside the one-electron reduced complex 1”. It is very unusual when one electron is delocalized over two distant from each other fragments of a molecule. Could the authors clarify, whether it is a two-electron reduced complex 1 (by one electron on each BIAN ligand), or it is a truly one-electron reduced complex?

Authors’ reply: Figure 9 depicts the spin density distribution on [{Ag(dpp-bian)}2Mo8O26]3-• anion, which is one-electron reduced with respect to [{Ag(dpp-bian)}2Mo8O26]2- anion. To avoid misunderstanding, the figure’s caption has been rephrased.
